# Befriending Your Food: Pigs and People Coming of Age in the Anthropocene

**Mary Trachsel**

Rhetoric, University of Iowa, Iowa City, IA 52242, USA; mary-trachsel@uiowa.edu

**Abstract:** Geologists and ecologists report that Earth is undergoing its sixth massive extinction event, an occasion that calls for radical revision of conservation ethics. The biologist Edward O. Wilson has proposed that conservation projects in the Anthropocene should be grounded in *biophilia*, an evolved, relational (or biocentric) mode of perception that activates aesthetic and affective responses to non-human life alongside cognitive understanding. Because biophilia includes non-rational modes of perception, the nurturing of biophilic conservation ethics cannot fall to ecology alone; imaginative literature, for example, can prompt readers to imagine and work to realize more environmentally friendly roles for humans and, further, can assist in cultivating a conservation ethic suited to current ecological conditions. In particular, coming-of-age novels about friendships between people and pigs offer an alternative to the industrial "pork story" that seeks to gain narrative control of relational norms between people and pigs, at the expense of biodiversity and ecological health. Three such novels published in 2017 depict human–pig friendships, a relational model created by pigs' shift in status from food to companion animals. In presenting this realignment, the stories facilitate development of a biophilic conservation ethic.

**Keywords:** biocentrism; biophilia; conservation ethics; friendship; human–animal relationships; narrative ethics; pig; relational ethics; young adult literature

---

## 1. Biophilia Opens a Relational Approach to Conservation Ethics

Well before the term "Anthropocene" entered public usage, scientists were already describing the end of the 20th century in geological terms, as an era witnessing Earth's sixth massive wave of extinction (See Leakey and Lewin 1995; Kolbert 2014). They report that an event like this has not registered in the geological record since the dinosaurs closed down the cretaceous period, 65 million years ago. The biologist Edward O. Wilson offers the name "Eremocene," the age of loneliness, as an alternative label that calls attention not to humans but to other forms of life that are rapidly becoming lost to the world. Such reports rarely penetrate human awareness very deeply, given that we experience our lifetimes in months, years, and decades rather than geological eras, and that the human species itself apparently is not facing imminent extinction. Still, ecologists tell us we should be worried about the fates of our fellow species, because our lives are tied up in theirs—because any one species' chances of survival are enhanced by the biodiversity of its environment, while its likelihood of extinction rises as fewer species share the work of shoring up the system against ecological collapse.

Like many other biologists and environmentalists[1], Wilson has argued that we need to take a long view of ourselves as participants, along with all other species, in a global web of life in order to establish

---

[1] Many environmental scientists and philosophers throughout the world share Wilson's premise that human action to preserve the non-human environment requires a departure from anthropocentric ethics and an engagement of multiple human perceptive modes. A listing of but a few of these thinkers and writers includes the Norwegian philosopher Arne Naess, who coined the term and developed the concept of "deep ecology," the US science writer, Rachel Carson, whose *Silent Spring*

a conservation ethic that is scientifically grounded and more fully engaged with non-human nature than science alone permits. Wilson envisions a coming together of biology and bioethics to produce "moral reasoning of a new and more powerful kind" (*Biophilia*, p. 138), a multi-modal, *relational* way of knowing that he calls "biophilia." Literally, the term translates as "love for life," but Wilson means it much more broadly. For him, biophilia describes an orientation to the natural world that engages not just the sensory and rational intelligence of the scientist, but also the aesthetic and affective powers of human mind that cause us *sapiens* to *care* about what we know of life beyond ourselves.

Biophilic perception, Wilson has hypothesized, arises from an innate sense of human "affiliation" with other forms of life (Wilson 2002; Wilson 1984, p. 1). This relational way of knowing finds its biological explanation in the Darwinian premise of the phylogenetic continuity of all life on Earth. As a biologist, Wilson assumes a human kinship with other living things that is traceable in our common genetic ancestry. To the extent that we regard our own species as the crown of creation and the mind or soul of the planet, we are disinclined to acknowledge nonhuman kin, but Wilson rejects this anthropocentric worldview. In its place, he proposes a conservation ethic in which interspecies kinship neither shames nor diminishes humanity, but instead, elevates nonhuman life to family status. Such a move, Wilson believes, is necessary to dissuade us from treating other species as though they are disposable. At the heart of Wilson's biophilic conservation ethic is a human willingness to share Earth's finite natural resources with other species, and in performing this willingness, he recommends a property-sharing contract that dedicates half the Earth's surface to the support of nonhuman life (Wilson 2016).

Wilson worried in 1984 that bioethics was lagging behind biological understanding of human continuity with the rest of life. As a discipline, bioethics has been primarily concerned with human-centered questions about the interface of human life and technology—about the allocation of organs for transplantation, for example, or about the manipulation of human genetics and reproduction, or about the technological capacity to extend or terminate human life. Alongside these anthropocentric concerns, Wilson noted, bioethicists at the end of the century had "only begun to consider the relationships between human beings and [other] organisms with the same rigor" (*Biophilia*, p. 119). Biophilia is his response to this imbalance, a biocentric morality that arises from feelings about, as well as knowledge of, our connectedness to other-than-human life.

Wilson has argued that science education shares with other academic branches a moral educational mission to promote care as well as knowledge of the natural world. Science, he maintains, should help humanity compose an ethical response to the ongoing loss of biodiversity around us, and this response should be informed by instincts and feelings excluded by the calculational reasoning and empirical focus of science. Many ecologically-minded science education programs throughout the world do seek to foster feelings of care that translate into action on behalf of nonhuman life. Jane Goodall's internationally recognized "Roots and Shoots" program is a prominent case in point, and in the U.S., science education with an emphasis on relational knowing has assumed the battle cry, "No Child Left Inside." A program directly influenced by Wilson's biophilia hypothesis is Richard Louv's (2005) Children and Nature Network, popularized through the best-selling *Last Child in the Woods* and *The Nature Principle* (Louv 2011). As in comparable science education programs, the methodology of Louv's "New Nature Movement," is immersion in the natural world to counteract "nature deficit disorder" and to re-establish the aesthetic and affective bonds with nature that have evolved within the human psyche.

---

(1962) brought public awareness to the environmental impacts of human activity, the Australian philosopher Val Plumwood, whose *Environmental Culture* (2002) lays out the framework for "ecological rationality," and the US environmentalist Frances Moore Lappe, whose *EcoMind* (2011) similarly prescribes an internal transformation in order to bring humans into greater harmony with the rest of the natural world.

A shortcoming of immersion programs like Goodall's and Louv's, however, is that their implementation is largely confined to children's education.[2] Although Louv argues in *The Nature Principle* that adults as well as children suffer from nature deficit disorder, most advanced levels of science education continue to focus exclusively on scientific method, empirical knowledge, and the use of observational enhancement technology. Advanced science education continues to separate the biological science of ecology from the relational experience of nature, reinforcing a conceptual divide between natural science and naturalism. In such a system, the notion of human kinship with other organisms appears sentimental and non-scientific, discouraging the revolution in moral reasoning that Wilson and others have insisted we need.

Cultivating a sense of kinship with nonhuman life, then, especially as children age into adulthood, becomes a moral education responsibility that science alone is not equipped to uphold. A scientific view of more-than-human nature exercises and sharpens human observational powers, but it does little to cultivate, and is often hostile to, narrative engagement. By "narrative engagement" I mean a relational engagement that entails *caring* about characters to whom things happen.

Many nonfiction accounts of interspecies relationships perform the narrative function of cultivating care for and about the nonhuman world. Ethologists in the narrative footsteps of Konrad Lorenz, for instance, have introduced *anecdotes* about nonhuman individuals to the observational repertoire of science, as have field primatologists, whose narrative styles were pioneered by Kinzi Imanishi in Japan and by Jane Goodall, Dian Fossey, and Birute Galdikas in Europe and the U.S. The interspecies narrative focus of the present study, however, is imaginative literature—fictional stories about relationships between human and nonhuman characters. In particular, I focus on the fictional type known as the coming-of-age story—what the German language has famously named *der Bildungsroman*—the story of a young person's moral crossing from childhood into maturity. This kind of story holds considerable appeal for listeners and readers who are themselves undertaking or preparing to undertake such a crossing, and it has historically served as a powerful medium of moral education. The present study examines an intersection of the coming-of-age story with another narrative form of moral instruction, what the moral philosopher Margaret Urban Walker (2007) calls "narratives of relationship." Walker's primary concern is human-to-human relational morality, but this study applies her moral analysis to a particular type of interspecies narrative of relationship: Stories about friendships between people and pigs.

In scrutinizing this very particular genre—coming-of-age stories about human–pig friendships—as a text of moral education, I trace a development in a relational worldview that supports a biophilic conservation ethic. These stories perform this support by recategorizing a quintessential food animal as a companion animal. In *Biophilia*, Wilson observes the human tendency to "favor certain animals because they fill the superficial role of surrogate kin," noting that dogs are especially apt to be chosen for this position because they share "humanlike rituals of greeting and subservience" that have earned them a reputation as man's best friend (p. 126). On the surface, pigs are improbable candidates for friendship with humans, given their long history as a domesticated food source. Today, however, as the massive scale of pork production around the world steadily hastens a global loss of biodiversity and threatens public health, ethical reassessment of our human relationships to pigs has become an urgent matter. In imagining pigs as friends instead of food, coming-of-age novels function beyond mere entertainment, as moral instruction that advances a biocentric ethic by challenging the human-centered morality of pig domestication and pork consumption.

My own geographical and cultural perspective comes from the U.S. heartland, specifically my home state of Iowa, where I came of age on a farm in a vast agricultural expanse punctuated by once-thriving small towns. From the 1950s through the 1970s, when I was growing up, pigs were

---

2   Other examples of immersive biology education for children include Joseph Cornell's Sharing Nature Foundation, presented in Cornell's many books, including *Sharing Nature with Children* (1979) and *Sharing Nature with Children II* (1989), and Gary Paul Nabhan and Stephen Trimble's *The Geography of Childhood: Why Children Need Wild Places* (1994).

part of the landscape. We kept ours in a lot, a pasture, a barn, and a hog house to the east and south of our own house. Today, although the number of pigs alive in Iowa has topped 23.7 million, and Iowa is by far the lead contributor to the new record, 73.5 million pigs living in the U.S. at any given moment (Associated Press 2018), you can drive across the state on country roads and never see a pig. Nearly all of Iowa's pigs are confined in concentrated animal feeding operations (CAFOs), and the process of breeding, feeding, and slaughtering pigs and packing and shipping pork is one of the few mainstays of our state's economy. Iowa has an especially large and concentrated pig population, but comparable conditions prevail in many spots across the globe. China tops the list of countries by national herd size, followed by the European Union, the United States, Brazil, and Russia (Global Mega Producer). The living conditions of the vast majority of domesticated pigs around the world are remarkably similar, as industrialization has imposed on pig farming the standardizing imperatives of mass production. Despite the geographical and cultural limitations of my own personal and academic standpoint, then, the problems that concern me in this article are global. In my very precise selection of narrative types to study, I necessarily overlook differences in human valuations of pigs across cultures, and changes in these valuations over time. For Jewish and Muslim cultures, for instance, pigs have never occupied the category of food animals, being classified instead as unclean beings that are both inedible and unfit for human companionship. In these cultures, pigs have been categorized as something akin to vermin. This despised status resonates with pigs' position in the Western culture that has produced the novels analyzed here. As nouns in the English language, "pig" and "hog" (along with "swine") are derogatory terms that describe people as greedy, fat, slovenly, and brutish, while as verbs they denote selfish, sloppy, and excessive consumption. These extremely negative associations coupled with the fast-growing, global spread of industrial hog farming conditions that keep pigs out of sight and out of mind spotlight the radical transformation of pig–people relationships advanced by the farmed animal sanctuary movement in the US and elsewhere and by the US coming-of-age narratives of relationship examined in this article. Mizelle's (2011) *Pig* gives a more comprehensive account of humanity's often conflicted relationships with pigs across cultures and over time. Prized for their meat and hides, favored for their ease of upkeep, despised for their ugly appearance and destructive habits, and sometimes cherished for their intelligence and personable dispositions, pigs resist easy or universal classification by humans. For this reason, they present a significant challenge to the necessary task of realigning human relationships with the rest of nature in the present age of environmental crisis.

## 2. Coming of Age as a Relational Process

A *Bildungsroman* tells of moral development through life experiences that transport a character from childhood to maturity. In many coming-of-age stories, these transformative learning experiences are relationships that a central character must choose to continue or terminate. Tragic romances like the tale of young King Arthur, for example, present growing up as a process of severing ties, resulting in the loneliness of lost relationships. A picaresque, like *Tom Jones* (1749), presents the path to adulthood as a story about making and breaking relationships with ill-chosen (but fun) companions. Romantic comedy likewise depicts coming-of-age as a process of learning to distinguish right from wrong relationships on a path to perfect partnership. Across all these genres, the coming-of-age story assumes an overarching narrative form that the moral philosopher Margaret Urban Walker (2007) refers to as a "narrative of relationship"—a biography of *them* or an autobiography of *us.* Walker maintains that narratives of relationship are one of the fundamental building blocks of moral understanding. Examining self-development in social relationships with others, she explains, illuminates "the acquired content and developed expectations" of relational identity and "the type of trust" in which relationships are rooted. Thus, probing the essential links between self and other, narratives of relationship prompt storytellers and listeners alike to ponder a relationship's "possibilities for . . . continuation," (p. 117) including, of course, its possibilities for discontinuation.

Many relational coming-of-age narratives spotlight young protagonists' fraught relationships with authority figures, who range from older siblings and parents to teachers, clerics, and representatives of the law. Others examine the peer relationships that mediate a young person's moral development, as in John Knowles' *A Separate Peace* (1959), a young-adult friendship story of betrayal and regret. A special subset of the coming-of-age friendship story features young people in relationships with non-human animals.

Interspecies relationship stories are common in children's literature, which freely fashions human friendship with a wide variety of species, including toy animals, as in A.A. Milne's *Winnie the Pooh* or Margaret Williams' *The Velveteen Rabbit* (1922). By contrast, stories for young-adult readers tend to tighten the range of nonhuman friends to popular companion animals, the foremost being dogs and horses. Notable examples are Enid Bagnold's *National Velvet* (1935), John Steinbeck's *The Red Pony* (1937), Eric Knight's *Lassie Come Home* (1940), Mary O'Hara's *My Friend Flicka* (1941), Fred Gibson's *Old Yeller* (1957), and William Armstrong's *Sounder* (1969). Some exceptions to this rule extend the bond of interspecies friendship to non-domesticated species; Marjorie Rawlings' *The Yearling* (1938), for instance, tells of a boy's friendship with a deer, and Sterling North's *Rascal* (1963) is about a boy and a raccoon. A different type of exception, and the focus of my study, features friendships between young people and livestock animals, and in particular, pigs.

A species commonly derided for gluttony and filth, pigs are surprisingly popular friends in stories for children and maintain a vigorous presence in young adult stories as well. White's *Charlotte's Web* (White 1952) exemplifies the distinctive feature of children's stories of this type: The pigs miraculously escape a tragic ending as edible livestock. One of the central characters, a pig named Wilbur, first appears as the infant ward of an 8-year-old human rescuer, Fern. As the plot unfolds, Wilbur's fate migrates from dependence on Fern to friendship with Charlotte the spider, who engineers the miracle of Wilbur's permanent escape from the livestock category. Wilbur's story further illustrates a popular feature of friendship stories about pigs, their beginning in a pink-and-white infancy that calls to mind Caucasian human babies. This is the impression Wilbur, in *Charlotte's Web*, initially makes on Fern when she opens the box and sees him for the first time: "There inside, looking up at her, was the newborn pig. It was a white one. The morning sun shone through its ears, turning them pink" (p. 4). From this moment, Fern adopts Wilbur as her baby and mothers him until he is too big to live in the family's house or yard, at which point she sells him (as a commodity) to her neighboring uncle.

Porcine biographies such as Sy Montgomery's *The Good*, *Good Pig* (Montgomery 2016), Jenkins' and Walter's *Esther the Wonder Pig* (Jenkins and Walter 2016) and Matt Whyman's *The Unexpected Genius of Pigs* (Whyman 2018) demonstrate that in real life, pigs' social intelligence and trainability can lead to their prolonged friendship stories with humans, and these traits no doubt further contribute to pigs' popularity in fiction. With a flexible repertoire of social behaviors broadly overlapping those of established companion animals such as dogs, pigs can in fact actively solicit and engage human attention and build interspecies companionship. Pigs' reputation for intelligence and affability, coupled with their despised status among domesticated animals, further explains their popularity in coming-of-age stories, as their position surely resonates with the miseries of adolescent alienation.

In real life, pigs stand out among livestock animals as creatures domesticated exclusively for slaughter. Sheep, goats, and llamas provide wool; sheep, goats, cows, and camels provide dairy; fowl produce eggs and down; horses, oxen, camels, and llamas are workers, but pigs are simply pork and leather. Most stories of relationships between people and pigs, therefore, unfold as tragedies of sacrifice and betrayal, a storyline that tests an interspecies morality framing domestication as "animal consent."[3] Budiansky (1992), an early proponent of this perspective, defines human–livestock animal relationships as the outcome of an ancient "covenant of the wild," by which, he explains, livestock species long ago made an "evolutionary choice" to affiliate with humans. According to Budiansky,

---

[3] For a discussion of "the myth of animal consent," see Foer (2009), pp. 99–101.

accepting the biological trade-off between the captivity and slaughter of individuals on the one hand, and the long-term survival and proliferation of the species on the other, livestock animals have in effect agreed to co-author the tragedy of their friendship story with humans. Honoring our side of this bargain, Budiansky argues, means accepting a relational morality such as Walker proposes—one grounded in a trust that one another's expectations will contribute to the terms of a relational identity. In Budiansky's words, "just as sheep may 'expect' to be killed . . . , they also impose moral demands on us because they . . . 'expect' to be better off than in the wild" (p. 167).

In the case of pigs and people, an ethics of animal consent requires that people consider the expectations of pigs in their domesticated captivity, the very subject examined in a recent spate of coming-of-age novels about friendships between young adults and pigs. All published in 2017, Jodi Kendall's *The Unlikely Story of a Pig in the City*, Paul Griffin's *Saving Marty*, and Corabel Shofner's *Almost Paradise* invite young adult readers to imagine individual pigs' expectations of their captive relationships with humans. The coming-of-age storyline shared by all three novels features a young person faced with decisions about how to continue or terminate a friendship with a pig. All three prompt imaginative speculation: What if pig slaughter were not an inevitability? What alternatives to tragedy might describe pigs' and people's relationship narratives? In response to these questions, all three stories reclassify pigs from food to companion animals, proposing a categorical shift that challenges the ethical norm of porcine consent to captivity and slaughter.[4]

Individual humans shifting individual animals from their "natural" destiny as human food to a radically different destiny as human companions neither stops nor perceptibly slows the growth of industrial pig farming and its massive environmental impact on the biosphere. But marketing data reveal that stories about pigs do affect the food choices of young consumers. Consumption of pork at fast food restaurants fell temporarily but significantly in the wake of movie releases of *Charlotte's Web* and *Babe* (O'Connor 1995; Nobis 2009), for instance. Therefore, I surmise that the stories about pigs that I study here may also somehow matter as a cultural force that nurtures an expression of biophilia, an attitude of caring about non-human beings in a new, more expansive way. Imagining pigs not as commodities but as participants in the pages and chapters of human biography, I contend, is a psychological exercise that cultivates biophilic ethics.

## 3. Coming to Terms with Slaughter

Pig-centered friendship stories for very young children generally avoid mentioning the terms of pigs' real-world domestication, while stories for older children tend to acknowledge the sad destiny of pigs, but only until fantasy intervenes to alter this fate and produce a happy ending. For young adults, however, the traditional pig–human friendship narrative equates adulthood with accepting the inevitability of pig slaughter. *The Born-again Carnivore*, a guide for small-scale farmers committed to humane meat production, endorses this tragic version of moral maturity when it warns against befriending pigs or regarding them as anything but food. "It's best not to give the animals names other than Freezer or Pork Chop," authors Sue Mellis and Barbara Davidson advise (Mellis and Davidson 1990, pp. 33–34). In the classic U.S. pig–human tragedy for young adults, Peck's (1972) *A Day No Pigs Would Die*, an autobiographical narrator fails to heed such advice but eventually crosses into adulthood when, for the sake of his family, he helps his father, a butcher, slaughter his beloved pig, Pinky.

The 2017 novels replace this tragic plotline with a story of lasting trust, if not continued association, between pigs and their young human friends. Collectively, they work to enlarge and diversify an anthropocentric worldview in which pig slaughter is unquestionably warranted by human carnism. By drawing attention to domesticated pigs' expectations of their human keepers, the stories assert an interspecies ethical perspective at a moment when the multi-billion-dollar pork-production industry is

---

4　On categorization of nonhuman animals, see Herzog's (2010) *Some We Love, Some We Hate, and Some We Eat* and Joy's (2010) *Why We Love Dogs, Eat Pigs and Wear Cows.*

intent on controlling the narrative of pig–people relationships with what it calls "the pork story" or "the industry story."[5] In this story, pigs' welfare expectations are explicitly defined as the living conditions that enable their swift and uncomplicated progress from infancy to death. Pork producers uphold the human side of the domestication covenant with pigs by providing these conditions, measuring their success against standards that quantifying pig-life as litter size, survival rate, time-to-market weight gain, and feed-to-meat ratios. In the industry's pork story, good care for pigs translates into human profit, while poor care leads to human loss. As a 2017 article in the U.S. labor and trade publication, *National Hog Farmer* asserts,

> [P]ig farming is all about selling pork. It is not about selling individual animals but raising delicious pork for the global table, and that all starts with the sow. The production and efficiency of the sow is quintessential to the success of the swine business. She has one job—to deliver the best pigs that perform well.

In response to animal welfare advocates' claims that pigs confined in CAFOs are worse off than they would be in the wild, the industry defends its production practices as "science-based" and justifiable by economic logic. As the Iowa Pork Producers Association declares on its website, modern production methods are "good for consumers, good for pigs," and "just good business."[6]

To support this anthropocentric interpretation of the covenant, pork industry marketing professionals use narrative strategically, often encouraging and coaching farmers on how to spread "the real story" of pigs and the people who know them best.[7] In the U.S., Pork Checkoff, a National Pork Producers Council product-promotion fund, advises farmers to use blogs, social media sites, and other outlets to broadcast "real pig-farming stories" premised on the human appetite for pork and depicting farmers as first responders to human need. Pork producers learn from marketing experts to promote and capitalize on food trends like "bacon-mania" in the U.S. and Canada and the entry of pork belly, lard, and chicharron into *haute cuisine* restaurants around the world. In the U.S. and perhaps elsewhere, the industry is particularly intent on spreading this message to an impressionable young-adult audience whose dietary choices may still be in flux. In a column advising hog farmers to think of their business as "feeding the masses," industry spokesman Kevin Schulz pondered the optimal age for planting the pork story in the minds of potential customers: "Some say junior high is when children are making their own decisions about what they are going to eat ... Still others say college students are a prime target due to all sorts of dietary messages being thrown their way" ("Feeding the Masses").

Emphasizing economic and dietary considerations in its pitch to young adults, the pork story circumvents questions about the relational expectations of livestock that are central to animal consent ethics. One way the industry dismisses concerns about pigs' expectations is by characterizing animal welfare advocates as outsiders who do not know "the real story" of pork production. It performs this move as the conditions of present-day pig farming actively discourage public attention to pigs' expectations by keeping pigs contained in CAFO buildings and slaughter houses to which public entry is prohibited by biosecurity regulations and "ag gag" laws that define efforts to expose the conditions

---

[5] A good example of this discourse can be found in the National Pork Board's (2014) *2015–2020 Strategic Plan: People, Pigs and Planet (2014)*. www.pork.org. The plan outlines an approach to the task of "telling the pork industry's story" and characterizes pork production as the activity of an industry that "has the best interests of consumers and pigs at heart."

[6] An example of how the pork industry calculates pig welfare is the research report by Cyril Roy, et al., "Determining Optimum Stocking Density in Nursing Pigs" in *National Hog Farmer*, 22 January 2018. The report acknowledges that "optimal space allowance increases productivity by maximizing feed intake and average daily gains of animals. However, optimum economic performance is influenced by high growth rates as well as by increasing the number of pigs per pen and overall barn throughput."

[7] See, for example, Kevin Schulz, "Food for Thought: Telling Pork's Story" *National Hog Farmer*, 24 October 2017; Hora, Greg, "Sharing Our Responsible Pork Production Story on the 2018 Heartland Tour," *Pork Checkoff Blog* https://www.pork.org/blog/shaing-responsible-pork-production-story-2018-heartland-tour/; Cheryl Day, "What is Your Excuse for not Telling Your Real Pig Farming Story?" *National Hog Farmer*, 17 October 2017

of pigs' living and dying as a form of domestic terrorism. Only occasionally do natural disasters such as tornados and floods abruptly reveal the conditions of pigs' present-day domestication to the general public and suggest the magnitude of this industrial enterprise by calling attention to pigs whose lives were contained within the pork production system. The recent young-adult pig–human friendship stories resist this erasure of pigs' expectations by substituting a narrative, relational morality in place of the contractual framework of pigs' domestication. These interspecies *Bildungsromane* thus open onto the larger questions Wilson has posed about the relational terms of humans' connection with and disconnection from other forms of life on planet Earth.

## 4. Fantasy Friendship Stories about Pigs for Children

"The Three Little Pigs" typifies pig stories for young children in overlooking any and all questions about the pigs' fate as domesticated animals. No version of the tale that I can recall speculates on how the pigs might taste, because a wild predator—the big, bad wolf—obscures any threat to the pigs from human predation. In fact, the pigs in the story stand in for humans as *sapiens*, thinkers whose superior wit enables them to dispatch the wolf. The especially clever and generous third little pig in particular turns the balance of predator and prey against the wolf, engineering a story ending in which the wolf tumbles through the chimney and into a cauldron to be cooked, presumably so the little pigs can eat him.

In the early 20th century, a popular pig character who remained similarly free from concerns about human dietary choices was Freddy from Brooks' *Freddy the Pig* (Brooks 1927–1958) series of 26 adventure novels and a book of poetry (Hochschild 1994). Unlike the three little pigs, Freddy does interact with humans, but his relationships with them rarely exceed acquaintanceship. Most people in Freddy's adventure stories are mildly comic characters on the outer orbit of his friendship circle. Often, they earn laughs by expressing vague surprise but no disbelief at discovering that a pig can talk and perform other "human" activities. As a character, Freddy is something of a clown; lazy and comfort-loving, he is a supremely affable companion to the other barnyard animals, who are his closest friends. But like the pigs who outsmart the wolf, Freddy distinguishes himself with a shrewd intellect that enables him to plan, orchestrate, and successfully carry out his schemes, even when they transport him and his barnyard companions into human venues such as airplanes and detective agencies. Unlike George Orwell's *Animal Farm* (1945), which presents a dark picture of porcine intelligence, children's stories generally represent pigs as good-natured tricksters who use their minds to maximize their own pleasure and comfort without depriving or threatening others.

A 21st-century children's book series about a pig, *Mercy Watson* (DiCamillo 2005–2009), plants the pig character, Mercy, in the center of a human family. Her human "parents," Mr. and Mrs. Watson, dote on her and refer to her as "the porcine wonder." Like a human child she eats at their table, rides in their car, and even sleeps in their bed, and like Freddy, she is pleasure-seeking and comfort-loving; the prime motivation in her life is her insatiable taste for buttered toast. Mercy resembles Freddy in pursuing adventures and accomplishing heroic feats—though often her achievements are inadvertent and performed in a comically bumbling fashion. While her human parents infantilize Mercy as a darling or a pet, other human characters recognize her as friendship material. The Watsons' neighbors, two elderly sisters named Eugenia and Baby Lincoln, are human onlookers who often get involved in Mercy's exploits. Eugenia, the rigid and overbearing elder sister, disapproves of pigs in general and Mercy in particular, but Baby secretly holds that "Mercy is good company" (*Mercy Watson to the Rescue*, p. 27), as she often finds herself unexpectedly accompanying Mercy on adventures that Eugenia would never permit. Firmly established as a pseudo-human companion and family member, Mercy is insulated from the individual sacrifices required of pigs by the domestication covenant.

The pig novels of the popular 20th-century British children's writer, Dick King-Smith, target slightly older readers than Mercy Watson's. The stories resemble *Charlotte's Web* and *Freddy the Pig* in their farmyard settings and domesticated animal communities, where pigs' social ties to fellow nonhuman animals, initially at least, take priority over their relationships with humans. King-Smith's

stories further resemble *Charlotte's Web* in raising the specter of livestock slaughter but dispelling it through fantasy. In Wilbur's case, Charlotte the spider miraculously intervenes in Wilbur's fate, persuading the humans who control his destiny that Wilbur has been divinely chosen for public adoration instead of consumption. Through Charlotte's machinations, Wilbur shifts his relational position with humans from livestock animal to permanent pensioner or pet in the Zuckerman's barnyard. In King-Smith's novels, by contrast, pigs are the engineers of their own categorical migration from livestock to companion animals.

The best-known of King-Smith's stories, *Babe, the Gallant Pig* (King-Smith 1983) became a popular film for children in 1995. Babe's story begins when Farmer Hoggett wins him as a tiny piglet in a contest at the fair. When Hoggett brings Babe home to the barn, the sheep dog, Fly, explains to her puppies that "the boss" will eventually eat the pig. This prompts a canine family discussion that reduces the differences between livestock and companion animals to distinctions between "stupid" and "clever" animals. When one of Fly's puppies asks anxiously if the boss will eat *them* when they grow up, his mother reassures her children, "People only eat stupid animals. Like sheep and cows and ducks and chickens. They don't eat clever ones like dogs" (p. 17). Babe, like other pig protagonists in King-Smith's novels, proves that pigs are at least as clever as dogs, implying that their intelligence entitles pigs to companion-animal rights and privileges. Babe successfully manages this shift in status by learning sheepherding skills from his foster mother Fly and from the sheep themselves. By the end of the novel, Babe is ultra-domesticated; the sheep trust him because he has polite manners, and Farmer Hoggett has "complete confidence" in his skills as a sheep-pig (p. 113).

Babe's public triumph at the sheepherding contest, like Wilbur's rescue by Charlotte, is a "miracle" (p. 113) to the human crowds who witness it, but to Farmer Hoggett, Babe's performance merely reaffirms the terms of companionable trust between human and nonhuman work partners. In the closing scene, Hoggett bends to scratch Babe between his ears, at which point "he uttered those words that every handler always says to his working companion when the job is finally done. Perhaps no one else heard the words, but if they did, there was no doubting the truth of them. 'That'll do,' said Farmer Hoggett to his sheep-pig. 'That'll do'" (pp. 117–18).

In *Pigs Might Fly* (King-Smith 1980), published three years before *Babe*, King-Smith tells the story of Daggie Dogfoot, a plucky runt of the litter who manages, through his own wiles and pig-headed determination, to permanently escape classification as livestock. His story, like Babe's, unfolds in a barnyard setting; Daggie's mother and her friends refer to the farmer as "Pigman," while the ducks all know him as "Duckman," each species believing that the farmer exists solely to tend to the needs of their own kind. The sows do not question the culling of runts, referring obscurely to those unfortunate offspring as "taken away." Their understanding of weaning is similarly vague and complacent, as they talk of their piglets' ritual "crossing the yard." After this termination of their maternal relationship with their piglets, the sows look forward to relaxing in the special meadow they call "Resthaven" where they can recover from the rigors of motherhood before their next breeding session with the boar, who is known in the barnyard as the Squire.

Initially "taken away" but miraculously escaping and returning to his mother, Daggie gains a reputation among the farm workers as an escape artist who is not worth the trouble to catch and kill. Small and deformed, he stays with his mother in Resthaven, where he makes friends with ducks who teach him how to swim. Because his deformed feet make him a powerful swimmer, Daggie is eventually able to save Pigman from being washed away in a flash flood. Through this heroic deed, Daggie, like Wilbur and Babe, miraculously escapes the inevitable fate of livestock animals. After rescuing Pigman and summoning help, Daggie, who has been swept downstream, rides back to the farm suspended on a rope from a helicopter, affirming the title's incredible assertion that "pigs might fly" as well as swim. When Daggie returns to the farm, Pigman is reduced to tears of gratitude, and the final scene shows Daggie swimming happily in the pond at Resthaven while crows look down in admiration, acknowledging Daggie's successful transcendence of categorical limits through "the miraculous skills of the amazing swimming pig" (p. 158).

King-Smith's third pig novel, *Ace: The Very Important Pig* (King-Smith 1990), introduces Ace as a food animal. Farmer Tubbs, when he first encounters Ace, contemplates the newborn piglet's destiny as meat for humans: "you will never be full grown . . . I shall sell you and your brothers and sisters when you're eight weeks old, and a few months after that, you'll all be pork" (p. 6). Shortly thereafter, however, Farmer Tubbs begins to realize that Ace understands the speech of humans, and he will eventually discover that Ace also understands and converses with all the other animals on the farm. At first, Farmer Tubbs cannot decide if he is going crazy or witnessing a miracle, but eventually, like Hoggett, he grows confident that Ace understands every word he says, and he begins to accord Ace the privileges of a companion animal. Gradually, Ace wins the trust of Farmer Tubbs' cat and dog, and through these friendships with favored companion animals he gains access to the house, where the three animals like to gather on the sofa in the afternoon to watch their favorite television shows. Like the humans in *Freddy the Pig*, Farmer Tubbs is surprised but credulous when he discovers that Ace has learned how to read the tv programming guide and can operate the television set. Tubbs begins to invite Ace to accompany him on trips to town and even to drink with him in the local tavern.

By the end of the novel, Ace's companion-animal performance has earned him regional and then national fame; he accompanies Farmer Tubbs to London for a televised appearance that increases his fame and draws prospective buyers to offer Tubbs large sums of money for his amazing pig. By this time, however, Ace's position as a companion animal is firmly fixed, and the farmer claims Ace as a friend instead of a commodity. In the closing scene Tubbs muses to himself, "What good would any amount of money be to you if you had to part with him? Why you wouldn't have no one to watch *West Country Farming* with. You wouldn't have no one to keep you company in the old pickup. You wouldn't have no one to enjoy a drink with at the Bull" (pp. 133–34). Over the course of the novel, Ace successfully engineers a permanent relational shift from food to friend, even coming to assume the exclusive status of "best friend" to his human companion.

A final example of human–pig relationship stories for preadolescents is Briggs-Martin's *The Water Gift and the Pig of the Pig* (Briggs-Martin 2003), narrated by Isabel, the granddaughter of a retired sea captain who keeps as a companion the last pig of the last litter of the pig who once sailed with him around Cape Horn. Isabel describes the pig as doglike in her intelligence, and on that basis she, like Farmer Tubbs, accords the pig exclusive best-friend status: "She is smart enough to count to five. She follows me like a dog and sits in the front of the boat when Grandfather and I go fishing for haddock. I would say the Pig of the Pig is my best friend. We have a secret handshake" (np). But eventually the Pig of the Pig's position as friend and companion is threatened when she disappears and an evil neighbor, Ben Stinchfield, goes looking for her. "I knew Ben Stinchfield was thinking of bacon and ham, pork chops and pigskin gloves" (np), Isabel worries, knowing that outside her family circle, the Pig of the Pig defaults to livestock status. Determined to save her friend from this fate, the narrator rouses her grandfather to action and learns from him the mysterious "water gift" of dousing (finding underground water) with a forked stick. After the stick in Isabel's hand points to where the Pig of the Pig has fallen into a hole, Isabel and the pig reunite with their secret handshake and dance together, the pig happily restored to the category of friend.

## 5. Coming of Age as a Carnivore

*The Water Gift and the Pig of the Pig* anticipates coming-of-age stories about human–pig relationships in casting Isabel as a moral agent whose decisions and actions determine, at least in part, whether and how the human–pig relationship will continue or end. Like Isabel's story in which the Pig of the Pig is nothing but pork to Ben Stinchfield, coming-of-age versions of the pig–human friendship story explicitly acknowledge the relational limits imposed by pigs' livestock classification, but they no longer invoke fantasy to make this troubling reality miraculously disappear. As a result, the human protagonists in these stories must confront the moral problem of determining whether and how to continue or conclude the relational story that joins their lives with the lives of pigs. This problem

wrenches them from an anthropocentric view of the world by forcing them to consider the demands of other denizens of the more-than-human world.

In Peck's *A Day No Pigs Would Die* (Peck 1972), the narrator named Robert Peck ends his friendship with his pig Pinky by conceding her default status as livestock and performing his part in her slaughter. For Robert, the grim reality of adulthood is that he has to kill the thing he loves. His despair as he assists in Pinky's slaughter is the tragic culmination of his coming-of-age experience; he gains adulthood when he accepts that his friendship with Pinky is bound to an inexorable fate. Initially, Robert describes the friendship as one of mutual fondness and delight in one another's company:

> I'd had her just ten weeks and already she was about my size. I lay on my back on the grass so she could come up to me and I could see her face. It always looked to me like she was smiling. In fact I know she was. Lots of things smile, like a flower to the sun. And one thing sure, I knew that just like I could smile to see Pinky, she sure could smile to see me (p. 51).

For a while, Robert believes he will be able to continue his relationship with Pinky long into the future, as he intends to make her a breeder sow, producing pork rather than becoming pork herself. But when Pinky proves barren, this hope evaporates, and the relationship story proceeds inevitably to Pinky's slaughter and Robert's complicity in the deed. When Pinky first appears to Robert on her final day, she, like the Pig of the Pig, is doglike in her expression of affection, prompting Robert to comment, "People say pigs don't feel. And that they don't wag their tails. All I know is that Pinky sure knew who I was and her tail did too" (p. 136). Robert vainly seeks to rewrite the tragic ending of his relationship story with Pinky, silently and apologetically addressing his desperate hopes to her as he hugs her neck and breathes in the smell of her body: " ... try and understand. If there was any other way. If only Papa had got a deer this fall. Or if I was old enough to earn money. If only ... " (p. 136). But despite his reluctance, Robert complies when his father tells him, "It's time" (p. 136).

After holding the stunned Pinky down in the snow so his father can slit her throat, Robert watches Pinky bleed to death at his feet, then helps his father hook her jaws and haul her to a cauldron of boiling water. Robert's father makes it clear that accepting Pinky's true identity as pork is the definitive sign of Robert's manhood. Robert describes how, after scraping Pinky's hide and beginning to carve her into meat, his father paused to acknowledge his coming of age:

> At last he stopped, pushing me away from the pork and turning me around so as my back was to it. He stood close by, facing me, and his whole body was wet with work. I couldn't help it. I started thinking about Pinky. My sweet big clean white Pinky who followed me all over. She was the only thing I ever really owned. The only thing I could point to and say ... *mine.* But now there was no Pinky. Just a sopping wet lake of red slush. So I cried.
>
> "Oh, Papa. My heart's broke."
>
> "So is mine," said Papa. "But I'm thankful you're a man."
>
> I just broke down, and Papa let me cry it all out. I just sobbed and sobbed with my head up toward the sky and my eyes closed, hoping God would hear it.
>
> "That's what being a man is all about, boy. It's just doing what's got to be done."
>
> (pp. 138–39)

*A Day No Pigs Would Die* has been widely censored in the U.S. as unsuitable for young adult readers. Most charges that put it on several American Library Association's (American Library Association 1990–1999) most-banned-book lists in the 1980s and 1990s target the novel's raw depiction of farm life, often citing Robert's intervention in the problematic birthing of a calf, a staged fight to the death between a dog and a weasel, and a brutal and unsuccessful mating episode that leaves Pinky bruised and bleeding. Objections that the breeding scene between Pinky and Samson the boar connotes the human crime of rape were especially troubling to parents and educators, as some of the book's detractors feared the description of pig breeding was so explicit that young female readers, encouraged

to consider Pinky's feelings and empathize with her experience, might internalize and someday re-experience the scene as a traumatizing flashback. Critics did not particularly object, however, to the underlying ethics of pig slaughter that traumatize the narrator when he has to kill his friend. Pinky's slaughter, the novel suggests, is consistent with an ethical acceptance of humanity's place in a naturally occurring food chain of predator and prey. Reminders of this chain occur throughout the novel, when Robert hears a rabbit's death cry in the talons of a hawk, for instance, and when he shoots and dresses a squirrel for his family to eat. Robert's full and final acceptance of the covenant that legitimizes human carnivory is his ticket to adult status in his family and community.

## 6. Friends Don't Eat Friends

In his 2007 study of human relationships with nonhuman animals, *Brutal: Manhood and the Exploitation of Animals*, Brian Luke has argued that a coming-of-age course of action such as Robert's is born of a concept of masculinity distorted by sexism. Certainly, the ethical framework of the domestication covenant upholds a similarly masculinized ethic of exploitation, valorizing the expectations of humans by presuming non-human "consent" as a justification for violence and self-gratification. Indeed, it has long been noted that the scientific perspective participates in this distorted worldview, positioning Nature as a feminized "other" to be conquered and forced to reveal her secrets. This masculinized anthropocentric worldview trivializes and dismisses the affective and aesthetic ways of knowing nonhuman nature that Wilson seeks to restore with a biophilic conservation ethic. As numerous feminist critiques of science since the 1970s and 1980s have argued, the interests and methodologies of science often replicate the patriarchal value of male dominance by policing the boundaries between "science" and "not science," and by validating masculine epistemic norms of distance, objectivity, abstraction, and rationality, while excluding feminized ways of knowing such as intuition and emotionality (Arditte 1980). When it comes to animals, the assumption of human domination of nature justifies an instrumentalist approach that converts laboratory animals to pieces of scientific apparatus and enforces objective distance between scientist and subject (Bowling and Brian 1985; Fee 1982; Fausto-Sterling 1981; Leiss [1972] 1994).

The 2017 updates on the coming-of-age narratives about young people's friendship with pigs reject the logic of domination and dismantle the ethical framework of animal consent and human necessity that undergirds the plot of *A Day No Pigs Would Die*. All three novels resemble *Charlotte's Web* in featuring pigs who are rescued in their infancy from a pre-ordained, domesticated fate, and like Wilbur and Pinkie, the pigs in these novels bear a likeness to human babies; when Josie first lays eyes on Hamlet in *The Unlikely Story of a Pig in the City*, she describes the piglet as "all pink and squirmy and perfect" (Kendall 2017, p. 1). But while Fern Arable is an eight-year-old child when she stays her father's ax-wielding hand to rescue Wilbur, the protagonists in *Almost Paradise*, *Saving Marty,* and *The Unlikely Story of a Pig in the City* are troubled teenagers who must eventually assume the weight of responsibility for the lives of fast-growing hogs. In the process, all of them come to depend on their pigs for moral and emotional support, and in the context of this friendship they learn to consider their pigs' moral and emotional expectations. Fourteen-year-old Lorenzo in *Saving Marty* reflects that learning to respond to his pig Marty's expectations endows his life with purpose. Naming his pig after his absent father, a decorated soldier whose suicide Lorenzo learns about and comes to accept as he is caring for his pig, Lorenzo enters into a relationship that focuses his attention beyond himself and his human family and friends. His relational identity makes his growing-up experience a process of coming to recognize himself not just as a responsible member of the human family but as a participant in a much vaster kinship network with other forms of life. When the pig, Marty, seemingly empathizing with Lorenzo's teen-aged confusion and sadness, rests his head on Lorenzo's shoulder, Lorenzo has trouble distinguishing himself from his friendship with Marty, and wonders, "if I could still be me without him there every day" (Griffin 2017, p. 162). He credits this relationship with creating "a new me, the one Marty made me into" and explains that he "needed" to learn how protecting and caring for another could nurture the self (p. 162).

Josie, of *The Unlikely Story of a Pig in the City*, (Shofner 2017) feels lost in her large and financially insecure family, and she fears her recent growth spurt will exclude her from gymnastics, the single arena where she feels she can make her mark. As Hamlet, the baby pig rescued by her older brother, grows toward full-sized adulthood and the family's economic problems worsen, Josie comes to realize that the city is no home for a pig, however much comfort and emotional support she may provide. Assisted by her closest friends, Josie accepts the responsibility of finding a "forever home" for Hamlet, in the process developing self-confidence and learning to appreciate her family and neighbors. Thanks to Josie and her friends, Hamlet, like Marty, is successfully rehomed in a safe and welcoming place. While Marty moves to a farm sanctuary, Hamlet makes her new home on a small hobby farm, and both pigs' relocations are close enough to their former homes that their friendships with their people can continue through frequent visits.

Like Lorenzo, Ruby Clyde in *Almost Paradise* is fatherless, and her irresponsible mother requires more care from Ruby Clyde than she can provide in return. When the story begins, Ruby Clyde and her mother are traveling in a camper, pursuing a get-rich scheme of her mother's unscrupulous and controlling boyfriend, known to Ruby Clyde as "the Catfish." After the three travelers steal Bunny, a young pig in a trained-animal circus in Hot Springs, Arkansas, the Catfish fumbles an armed robbery attempt. When he and Ruby Clyde's mother are arrested and taken into custody, Ruby Clyde and Bunny escape the police by hiding in the bushes. Like Marty and Hamlet, Bunny begins to accompany his human companion on walks, sleeps and snuggles with her in bed, and becomes a critical emotional support system in her troubled life. As she works through her complicated family problems, Ruby Clyde takes responsibility for finding Bunny a safe, comfortable, and permanent home with the kindly owner of the truck stop the Catfish once attempted to rob. Unlike Lorenzo and Josie, Ruby Clyde does not anticipate a continuing relationship with her pig at the end of the novel, but she feels secure in knowing she has satisfied his expectations by providing a safe and happy home where he can live out his natural life. She reports of their relational story's ending, "And that is how I left my pig. Under the big eye that had seen that whole part of my life, one that I would shortly leave behind. Bunny. Living at the Red Eye Truck Stop, where he would spend the rest of his days lolling in the mud by Frank's door and winking at cowboys who stopped for gas or food. No Cadillac, no bacon. Life had worked out for Circus God's pig" (p. 267).

The transformation of tragedy to happily-ever-after endings occurs in all of these novels when the young protagonists deliberately resist the overarching pork narrative of their time. In all three novels, the pigs' young caregivers are, like Isabel in *The Water Gift and the Pig of the Pig*, acutely aware that the rest of the world regards their friends as food. In *Saving Marty*, Lorenzo's mother defers to the pork story when she first sees her son cradling the baby pig and asks, "You know that's livestock, right?" (p. 23). Later, when she sees the relationship developing further between Lorenzo and Marty, she implicitly invokes this story again as a reminder of real-world conditions: "A *pig* for a *friend*?" (p. 53). At the story's outset, Lorenzo too accepts the pork story as the inevitable overarching narrative of his relationship with Marty. From the beginning of this relationship, he anticipates its sad ending: "Already I knew how bad it would hurt, the day he'd take that last walk up the ramp into Mom's truck for the auction" (pp. 16–17). Ruby Clyde in *Almost Paradise* similarly ponders a dismal future for Bunny and identifies with his hopeless story. With her father dead and her mother in jail, she sees herself, like Bunny, as a "stray" and muses, "If they put stray girls in the state orphanage, what did they do to stray pigs? Bacon, that's what" (p. 77). Josie, in *The Unlikely Story of a Pig in the City*, immediately sees a need to protect Hamlet from her own family's appetite. Surveying her parents and siblings at the breakfast table amid the smell of bacon the morning Hamlet arrives, Josie thinks, "If it were up to them, Hamlet would be *ham*—and that was her destiny if I didn't stand up for her" (p. 13). Lorenzo has similar worries about Marty's fate when the pig goes missing and is captured by a neighbor who, like Ben Stinchfield, regards pigs only as human food: "'He was roadside, no tags,' Mr. Taylor said. 'Finder's keepers. I'll send you some steaks'" (Griffin 2017, p. 93).

The young humans in these stories all attain adulthood by preserving mutual trust in their friendships with pigs. Within these friendships, the pigs are active agents too, not just offering companionship, entertainment, and emotional support, but taking active steps to sustain and deepen the trust, thereby placing relational demands on their young human friends. Of the three pigs, Marty plays the most active role in building a friendship with his human; near the end of the novel, he saves Lorenzo from an attacking dog (*Saving Marty*, p. 179). The other pig characters contribute to their relationships more subtly. At a low point in Josie's life, for example, Hamlet expresses empathy and follows through with action: "Hamlet climbed out of the basket and snuggled up next to me like she knew I needed a friend right now" (*Unlikely Story*, p. 184). Ruby Clyde takes similar pleasure and comfort in Bunny's responsiveness to her feelings, and early on, when she and Bunny are hiding from the police, she suspects the pig has dragged her into the bushes with his makeshift leash to protect her from capture by the police.

The human–pig friendships in these novels thus prompt the human friends to ponder what it means for the pigs to be "better off" in the care of humans. Josie, deciphering Hamlet's wordless expectation of greater freedom than a city apartment can afford, decides that her responsibility is not, as she initially believed, to figure out how to keep Hamlet with her, but to find Hamlet a better place to live. Ruby Clyde, in transferring Bunny's care to a friend, similarly makes a moral decision on the grounds that "I had to do what was best for my pig" (*Almost Paradise*, p. 268). Lorenzo, although he knows that "wanting to spend time with him . . . would stay strong in me for as long as he lived" (*Saving Marty*, p. 163), arranges for Marty to live apart from him, though close enough to continue the friendship.

Considering how their pig friends can be better off in human custody than on their own prompts the young protagonists in these stories to change their behavior along with their thinking. All of them credit their pigs with modeling a good-natured optimism that they begin to incorporate into their own human worldviews. Beyond this, friendship with a pig also leads both Ruby Clyde and Josie to question the morality of meat-eating in ways that had not occurred to them before. Early in her relationship with Bunny, for instance, a very hungry Ruby Clyde impulsively orders "Pancakes please, with bacon" and is immediately assailed by conscience: "I thought of Bunny and I got a stabbing pain. "Wait! No bacon" (*Almost Paradise*, p. 88). For Josie, in *The Unlikely Story of a Pig in the City*, friendship with a pig coincides with an experiment with vegetarianism.

The pig–people friendships in these novels, moreover, all involve an intimate physicality, as Robert's delight in Pinky's smell, her gaze, her smile, and her wagging tail in *A Day No Pigs Would Die* echoes through these more recent stories of pig–human friendship. When Ruby Clyde bids her final farewell to Bunny, for example, she seals the moment with an intimate embrace in which their interspecies identities seem to merge: "I kneeled down beside him and wrapped both arms around his neck. Our hearts beat together for a few brief moments, then I let go and stood up" (*Almost Paradise*, p. 266). Lorenzo, too, takes pleasure in the physicality of his friendship with Marty, and reflects on this as a transcendent experience: "I'd miss scratching the fur on the side of his neck, his chest. I'd feel his heart in my fingertips, in my arms, rising into my spirit" (*Saving Marty*, p. 163). This type of physical intimacy, common in stories of human relationships with dogs, is rare beyond the infant stage in real-world relational narratives of pigs and people today. Outside of fiction, the overwhelming majority of these relationships unfold in conditions of CAFOs where the pig-to-human ratio may average approximately 3000:1 (Genoways 2014, p. 196).

Like Pinky and the Pig of the Pig, the pigs in these young-adult novels inhabit fictional worlds where a categorical shift from food to friend is justified by pigs' intellectual and behavioral resemblance to dogs. Marty and Hamlet both play with tennis balls and learn to sit, fetch, roll over, play dead, and perform other tricks commonly associated with dogs. Like Babe, Marty even breaks the companion-species barrier by excelling in a competition intended for dogs—in this case, a dog race rather than a herding contest. In all three novels, the pigs sleep in the beds of their human companions and accompany them on outdoor rambles. Bunny, for instance, crosses into canine territory by keeping

close to Ruby Clyde's heels and "trotting like a dog" (*Almost Paradise*, p. 123), an identity shift that other characters acknowledge by feeding Bunny peanut butter dog treats. Endowing pigs with qualities associated with dogs, the quintessential companion animal in most Western cultures, the novels naturalize the interpersonal friendships between young people and pigs, thereby facilitating pigs' escape from abstractions like "livestock" or "meat", to which Carol Adams (1998) attributes the moral erasure of food animals.

## 7. Pig Stories, Pork Stories, and the Narrative Universe of the Anthropocene

Publication of the novels in 2017 coincided with rising public doubts about the environmental sustainability of the CAFO model that steadily expands the environmental footprint of pig farming in the U.S. and elsewhere. That same year, Robert Gloss reported in *If Pigs Could Talk* (Gloss 2017) that a total of 115.4 million hogs were being slaughtered annually in the U.S., while the *National Hog Farmer*, a U.S. publication, reported that "all the major pork-producing countries want to grow their market footprint" (GlobalMegaProducer 2017). Simultaneously, scientists were attributing the unprecedented size of the hypoxic "dead zone" in the Gulf of Mexico to agricultural run-off from Midwestern U.S. states where hog-farming is most intensive (e.g., Eller 2018); public health groups were circulating concerns about hog CAFOs as potential breeding grounds for rapid swine flu virus evolution and transmission (e.g., Lantro et al. 2016); and citizens in leading pork-production states, North Carolina and Iowa, were bringing suits against large-scale pork producers for compromised air quality and declining property values near the facilities (e.g., Blythe 2018; Cooke 2018). In this discursive environment, young-adult friendship stories about pigs foreground a relational interspecies morality that challenges prevailing terms of our human covenant with the ultimate food animal: The domesticated pig.

As moral tales, these stories of interspecies friendship encourage readers to question certain human privileges in the domestication deal we have struck with pigs. When humans regard captive pigs through the relational lens of friendship, an entirely different set of expectations appear much more reasonable than when pigs are instrumentally defined as industrial farm products. Indeed, the interspecies relationships of the young-adult characters reveal to them the welfare expectations of pigs that Temple Grandin's research on environmental enrichment began to uncover in the mid 1980s, including demands for social bonds, intellectual stimulation, and the expression of innate behaviors such as rooting and nesting (e.g., Grandin 1988, 1989).

While animal welfare science exposes unmet expectations of pigs in CAFO settings and recommends ways to mitigate the suffering of pigs in these conditions, the 2017 novels mention factory farming only obliquely if at all. If the fictional pigs face an imminent threat of slaughter, it is individualized or only vaguely conceived. Marty, when captured by ne'er-do-well neighbors who intend to kill and butcher him on the spot, has the closest brush with the meat market, but his story never mentions the industrial-sized norms of pig slaughter today, nor does it probe the living conditions of pigs in the industrial pork-production system. Josie, in *The Unlikely Story of a Pig in the City*, suspects that a man responding to her classified ad about a home for Hamlet is somehow connected to pork production, but she quickly blocks this development in Hamlet's biography by rejecting the man's offer.

Rather than mirroring the reality of today's pork industry, these novels, in selecting pigs as a particular embodiment of interspecies friend for readers to imagine, emphasize particular friendship virtues for emulation. The young humans who befriend pigs in these stories must rapidly adjust their own expectations of the friendship to accommodate the ever-increasing welfare demands of the pigs. They are all at some point in their stories nonplussed by their cute little pigs' rapidly growing size and appetite; they all struggle to meet their pigs' good-natured insistence on ever more food, space, and freedom; and they all choose to preserve the interspecies trust at the heart of the relationship on terms largely set by the pigs.

By rejecting the option to terminate the relationship as it gets harder to maintain, the young protagonists disrupt anthropocentric norms and incline to a biocentric worldview anchored in relational understandings of the other-than-human world. In honoring improbable interspecies friendships despite social and economic pressures to abandon them, they come of age by refusing to regard a non-human life as disposable. Choosing to continue the interspecies plural narration of "my pig and I" or "we," the characters affirm the affective and aesthetic interspecies ties of biophilia and carry them forward to adult understanding.

Persistence in interspecies friendships with beings that typically inspire revulsion and contempt is a different kind of virtue than persistence in friendships with conventional companion animals. It calls for a shifting of categories and a rearrangement of values, and it represents a step toward a deep-caring biophilic conservation ethic of the sort E.O. Wilson and others prescribe for this moment on the planet. These novels do not simply invite young readers to imagine an unlikely friendship story, they also invite them to imagine a world in which young people like themselves can choose to continue these friendship stories in defiance of their "inevitable" endings. In this, they anticipate the moral questions humanity must ponder in the time of the sixth great extinction on Earth—questions of whether and how we will continue in the company of other Earthlings.

I have argued that the books in this study contribute to a moral education project that current conditions demand. Massive losses of biodiversity across the Earth herald the dawning of the Age of Loneliness and illuminate the need for an ethical orientation that values interspecies relationships and prompts human action to preserve them. In befriending food animals and rejecting the inevitability of their slaughter and consumption, the young human protagonists demonstrate what one of my reviewers described as "a willingness to fall back into a natural order," an acceptance of humanity as a part of, rather than apart from, the rest of nature. To be sure, the books fail to address the enormous scope and reach of the CAFO system in which millions of characters like Marty, Hamlet, and Bunny complete their brief and "unnatural" lives. To introduce the real-world conditions of factory farming would dampen the happy ending of any of these stories by conjuring a backdrop of massive tragedy. Clearly, much work remains in the task of reintegrating humanity with the rest of nature throughout the world, but I contend that novels such as these are laying the groundwork for the work remaining. Moral philosophers Gruen and Chris (1997) assert that "exploring the positive political implications of our intimate relationships with companion animals" is an important extension of the feminist *credo*, "the personal is political." They argue that logics of detachment and domination (12) that govern human relationships with nature "obscure "a great deal of knowledge concerning our continuity with nature" (12). Contending, like Wilson, that environmentalism must transcend these rationales and engage other ways of knowing, they express concern that environmentalists in particular have inadequately credited the power of interspecies intimacy as a source of moral knowledge. They propose that interspecies friendships constitute an "ethical starting point" for bridging the moral distance that prevents us from caring what happens to other species. Interspecies friendships, they submit, enable "radical transformations of our orientations toward nature" (20) by providing access to biocentric moral reasoning that anthropocentrism so deftly excludes from the human worldview.

One way the pork industry maintains control of the pig—human narrative of relationship is by keeping pigs away from the human worldview. When humans are denied entry to slaughterhouses and CAFOs for whatever reasons—biosecurity, food safety, product quality control, or concerns about property damage and economic loss—pigs become what moral philosophers know as "distant others" for whom it is difficult to care. Imagining a real, live pig as a companion is as close as many young readers and video viewers today will ever get to interacting with life in its porcine form, but books like the 2017 coming-of-age novels help them make this approach. Imagining pigs as friends instead of food or vermin is an exercise that can have long-term effects if it finds reinforcement in the surrounding narrative environment. The industrial pork story is a powerful force in this environment, but its narrative monopoly is challenged by the texts I have presented and a host of others, including pig stories coming from a proliferation of farmed-animal sanctuaries, from the social media presence of

"celebrity pigs" like Esther the Wonder Pig and Priscilla, and from tourists in the Bahamas who tell stories about swimming with the "Pigs of Paradise." When, at last, it becomes clear that we cannot continue to live on this Earth while maintaining our present relationship with pigs, such alternative narratives of people and pigs together may persuade us that although we no longer eat them, pigs are companions we do not want to lose as we advance into the Age of Loneliness.

**Author Contributions:** Mary Trachsel was solely responsible for all research and writing.

**Funding:** This research received no external funding.

**Conflicts of Interest:** None.

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
