# Peer review of "Befriending Your Food: Pigs and People Coming of Age in the Anthropocene"

_socsci, doi:10.3390/socsci8040106_

Round 1
Reviewer 1 Report
Overall, this is a really fascinating piece. Reading about the history and the prevailing narratives about pigs and humans published in 2017 was quite fascinating especially with my history of working with pigs. Throughout the piece, I wanted to know how the 2017 publications compared to the previous years and how this has or has not coincided with cultural phenomena around pork eating (like the current bacon obsession). However, I recognize this is far outside the scope of this paper. I had a few suggestions, but nothing major. My only “big” critique is that the conclusion leaves the reader with a “so what?” feeling, so I would like to see the author tell us how this analysis could be applied. Could it be used to move writing about human and pig relationships into a new direction? Should we think and analyze these writings differently? What is missing from these texts that future writers need to consider? Perhaps, the author sees a different direction, which would be fine. It just needs an applied component.
1) (On page 5 lines 194-199) Have you thought about the current movement to place pigs in pasture or organic pig farming to support pest control, etc., which is presented as a more romanticized form of production, but still ends in the slaughter of pigs? Below is an interesting article about this that you might be interested in. It might fit into the current work or not. I’m not wedded to seeing it there, but it is an interesting perspective.
a. Intentional Shared Suffering: A Comparative Analysis of Varied Pig Production Methods in a University Setting
Cameron Whitley, Seven Mattes, Rachel Kelly
Animalia An Anthrozoology Journal
http://animaliajournal.com/2015/10/25/intentional-shared-suffering-a-comparative-analysis-of-varied-pig-production-methods-in-a-university-setting/
2) I found it interesting that you did not talk about the cultural interest and current trend of eating bacon on everything and how this may/may not impact pig/human relationships especially when talking about “emphasizing dietary over ethical choices…” (page 5 line 212). Again, this is not a critical issue, but something to consider.
3) I like how you say, “…present-day pig farming actively discourages attention to pigs’ expectations by keeping pigs out of sight…” This is completely true, but this narrative occasionally gets disrupted when climate induced storms make visible the invisible, but flooding these facilities and drawing in the empathetic responses of viewers.
4) (Page 12 line 553) You mention, “The 2017 novels mention factory farming only obliquely. If the fictional pigs face the threat of slaughter, it is individualized or only vaguely conceived.” Then you state (line 564) “these novels, by selecting pigs as a particular embodiment of interspecies friend for readers to imagine, emphasize particular friendship virtues for emulation.” I think this is a very important idea. While these novels seek to connect readers to pigs on an individual level, they (in some ways) may fail to address the systematic nature of what is happening to most pigs by perpetuating a romanticized view of pig life that is seldom enjoyed (unless the pig is in a sanctuary). I think the statement you make here could be stronger.
5) I agree that the protagonists disrupt anthropocentric norms, but the books seem to not challenge the large (CAFO) structural realities of anthropocentrism as it relates to pork production. This goes back to comment 4. The author addresses this slightly, but the language could be stronger.
6) The conclusion asserts that the protagonists are fighting against social and economic pressures, which I agree with, but I would also assert (as is demonstrated by Robert’s story and lots of research) that people are inherently drawn to and find relationships with animals, so the application of biophilia is really about leaning into the nature relationships that developed between species, not in constructing new values-as values (according to Shalom Schwartz are guiding principles in one’s life). It is a fight against external pressures, but a willingness to fall back into a natural order. Just something to consider. This does not necessarily need to be addressed in the paper.
7) I would have liked to see a better interweaving of biophilia throughout the paper and not just in the introduction and conclusion. I think this can easily be accomplished by adding a few additional sentences in each section that then remind the reader continuously of the theoretical application that is being positioned.
8) The only main issue I had was that, this piece lacks a “what next” in the conclusion. I wanted to know, what are the recommendations? What should scientists do? What should parents do to give kids more perspective? How should future writers approach writing about human and pig relationships? Basically, I wanted to know how this work could be applied and then be extended into other projects.
Author Response
Dear Reviewer #1,
Thank you for your very helpful review. Your "big" critique about the need for a stronger conclusion to address the "so what?" question was something other reviewers also brought to my attention, and I hope I have satisfied all three of you by thinking further on why it matters that these three novels offer an alternative moral universe to the one most of their readers inhabit, and what further work needs to be done to realize a conservation ethic that is grounded in something akin to biophilia. I live in Iowa, where it is blatantly obvious that human relationships with pigs have to change, as intensive corn, soy, and hog farming in the state have had devastating effects on biodiversity, so I have more explicitly situated my argument as originating from this particular perspective.
I read the article you suggested on "Intentional Shared Suffering," and although I haven't included it directly in my discussion, it will surely inform my future work on the subject of human-pig relationships, and I thank you for the reference. I have included some mention of the current craze for bacon and pork belly, something you can be sure the pork industry is backing. I remember reading their celebratory claim in a trade journal that "bacon is trending." And I have also included your very interesting point about how natural disasters (in Iowa it is not just floods but also tornados) sometimes dramatically expose the numbers and living conditions of farmed pigs to public view.
I have attempted to strengthen my argument about the significance of pigs in these coming of age stories by saying more about the point they conspicuously do not address: CAFOs are products of systemic anthropocentrism that immediately threaten ecological well-being I have identified the confrontation of this reality as a necessary next step in the process these novels initiate in their function as "starting points from which to close moral distance" (Gruen and Cuomo, 1997).
I have also tried to make biophilia and biocentrism more visible throughout the paper, but I still feel there are long stretches where, in keeping with my literary training, my focus on pig-fiction is unbroken. In the time I had for revision, I wasn't able to figure out how to relieve this relentless focus.
Finally, thank you for the phrase "willingness to fall back into a natural order," which is precisely what biophilia is, I think. Crediting you anonymously as a reviewer, I have used that phrase in my revision. Thank you for your careful attention to my manuscript.
Reviewer 2 Report
Thank you for the opportunity to review this paper, the content of which is at the fringe of my expertise, but completely aligned with my interest in animal-human relationships. I found the paper really interested and have provided an annotated PDF file with a small number of comments. As an objectivity- driven scientist please forgive me if any of the comments are too rigid in nature.
I thoroughly enjoyed this paper. There were a small number of places however where I felt that the concepts being discussed could have been described/explained in a little more detail to assist the reader's understanding.
The concluding section could also have linked more explicitly with the title of the paper.
Overall well written and generally very engaging.

Author Response
Dear Reviewer #2,
Thank you very much for your helpful review of my manuscript, "Befriending your Food." I was especially glad to receive feedback from a scientist, as my academic background and writing experience are deeply embedded in the humanities. I did not at all find your editorial suggestions "too rigid."
Your biggest concern--about the need for more development of the conclusion--was shared by my other reviewers as well, and I have considerably revised both the introductory chapter and the conclusion by more clearly defining key concepts (biophilia, biocentrism, conservation ethics) and linking them more directly to the discourse analysis I perform on books for young people about relationships with pigs. I have more explicitly identified my own position in the US state where pigs are most densely farmed and where environmental effects of prevailing pig-human relationships are impossible to overlook.
I hope my revisions to the manuscript have satisfactorily addressed your concerns.
Reviewer 3 Report
Dear Author,
Thank you for an interesting read that I think is worth publishing after reflecting on and finding answers to (some of) the observations I made while reading the manuscript:
This is clearly a manuscript written with a global orientation in mind. In that context it is probably worthwhile to take various religious / cultural ideas about pigs far more explicitly on board and wonder / explore how they might relate to taking especially pigs as examples for 'coming of age in the Anthropocene'?;
Related to the first bullet above, it seems there is some (white) 'US-centrism' in the manuscript. By taking American authors E.O. Wilson and Richard Louv as point of departure, it is probably worthwhile to explicitly denounce the suggestion that it might evoke that the US or 'white men' are 'guiding' or 'showing the way to' the global world here?;
Following the second bullet: On page 12 you write about 'this country' (541, italics added) without specifying that it is about the US (in relation to the number of pigs slaughtered on an annual basis);
To me the choice for especially pigs does not yet become clear enough in the text. The connection between 'biophilia' / 'nature deficit disorder' and the example of pigs / human-pig relations, to me, certainly do(es) not connect automatically to each other and therefore might need some more introduction / explanation?;
What also might need some explicit attention in the introduction is explaining the choice for fiction novels and not 'coming of age' stories in the more (auto)biographical genre, like for instance the works of Cynthia Moss on elephants, Gareth Patterson on lions, Robert Sapolsky on baboons, Roger Payne on whales, etc.?;
Page 9: Powerful example (!) of human-animal relations in the context of 'manhood' and masculinity: Might be worthwhile to theoretically / conceptually contextualise this example more explicitly in a gendered approach to biophilia and human-animal relations (see for inspiration for instance the excellent work of Brian Lukes (2007: 'Brutal. Manhood and the exploitation of animals')?
Talking about pigs specifically, I wondered if a reference to the role they play in George Orwell's moral tale 'Animal Farm' would be appropriate?
Looking forward to an updated version of the manuscript and wishing you good luck and inspiration in the process, kind regards.
Author Response
Dear Reviewer #3,
Thanks very much for your comprehensive and helpful comments and suggestions. You are right in saying that the manuscript has a global orientation, and I need to be more aware of a global audience. I have attempted to eliminate references such as "this country" and have been much more explicit in identifying my positionality--not simply in the U.S., but in the state in the U.S. where pigs are most densely concentrated in CAFOs. Although I didn't see room in this manuscript for discussing other cultural and religious ideas about pigs, I did acknowledge this global diversity and referred readers to Brett Mizelle's wonderful survey, Pig, which gives both a global and historical account of human-pig relations.
I also decided not to explicitly denounce possible readings of my manuscript as a suggestion that white men should lead the world, but I did position Wilson and Louv among the many other environmental philosophers and educators who promote a conservation ethic that engages affective and aesthetic perceptive channels along with scientific understanding of ecology. By including global representatives of this stance--e.g. the Norwegian philosopher Arne Naess, Jane Goodall, Rachel Carson, and Frances Moore Lappe--I hope to demonstrate that I do not intend to suggest that white American men have all the answers. I personally find Wilson's formulation of biophilia helpful because he presents it as a scientist who embraces more-than-scientific understanding of the natural world.
I have attempted to explain my choice of fictional accounts of pig-human relationships more clearly. Defining pigs as the quintessential food animal, much reviled for greed and sloth as well as meanness, and expanding my account of the moral education function of imaginative literature--particularly coming-of-age stories and narratives of relationship--I attempt to justify the very specific textual focus I've chosen for this particular study. In doing this, I have acknowledged that other narrative types also provide rich material for analysis--e.g. ethologists' anecdotes, primatologists' field research, and autobiographical accounts of living with animals.
Thank you for alerting me to Brian Lukes' study of masculinity and the exploitation of animals. I have included him in my reflections on the novel, A Day No Pigs Would Die. I have also included a brief mention of Animal Farm as a negative representation of pigs' intelligence, retaining the stereotype of pigs as greedy and mean.
I hope that the revised manuscript has satisfactorily addressed your concerns.
Round 2
Reviewer 3 Report
Dear author,
Thank you and it was a pleasure to read your text once more and see the improvements!
I have two final observations that I would like to share with you, not for self righteous reasons I hope, but for a continuation of a conversation:
* On page 3 and 4 you write: 'Despite the geographical and cultural limitations of my own personal and academic standpoint, then, the problems that concern me in this article are global. In my very precise selection of narrative types to study, I necessarily overlook vast differences in human valuations of pigs across cultures, and changes in these valuations over time. Brett Mizelle’s comprehensive Pig (2011) wonderfully provides the global perspective that this close-focus article does not'. You do not seem to return to this observation anywhere else in your text, which makes it a rather isolated statement with no real consequences?
* On page 11 you write: 'In his 2007 study of human relationships with nonhuman animals, Brutal: Manhood and the Exploitation of Animals, Brian Luke has argued that a coming-of-age course of action such as Robert’s is born of a concept of masculinity distorted by sexism'. This is where a reference to the importance of the concept of masculinity in human-animal relations starts and ends at the same time? As a consequence this reference remains rather isolated in your text (and therefore does not add much)?
Both of my observations boil down to two additions to your text that you probably wrote because of my earlier review, but didn't really 'take to heart', to become a more integral part of your text / argument. Maybe I can push that process a little more once again in this second review?
Good luck and best regards!
: ,
Author Response
Dear Reviewer 3, I apologize for taking some time to response. I was preparing for a conference trip and am now en route, but I hope to have some time soon to address your concerns. I have been thinking about how to respond and have a few thoughts and a question.
You're right in saying that I haven't substantively addressed your concern about culturally diverse relationships between pigs and people, largely because the CAFO model of hog farming has pretty much monopolized those relationships around the world in the last few decades. Right now, the largest pig population is in China, followed by the EU, the US, and Brazil. Conditions across these locations are largely the same. I could, of course, note that Jewish and Muslim cultures have never classified pigs as food animals, but as unclean and therefore inedible, and I might also mention that in some rural communities backyard pigs are still an important source of human nutrition. Because of pigs' intelligence and sociality, these more intimate domestic arrangements have certainly spawned (and continue to spawn) pig-people friendships not unlike the one in A Day No Pigs Would Die. Finally, I might mention the current and historical popularity of wild boars as a prey for hunters. Pigs quickly and easily go feral, and boar hunting is a popular sport in the US, especially in the southern states. In Texas, for instance, formerly domesticated pigs and their feral offspring are captive targets on ranches where hunters pay to hunt large game.
Pigs as prey for hunters certainly connects with your other concern about gendered relationships between pigs and people. The notion that coming of age means killing the (nonhuman) thing you love is a commonplace in coming of age literature for boys--e.g. Old Yeller and The Yearling--but not for girls. And, of course, the historical-cultural backdrop to all of this is the imperative to conquer and subdue non-human nature (and the scientific imperative to expose nature's secrets) as opposed to coexisting and allowing our human selves to "know" nature in emotional and aesthetic ways in addition to understanding it rationally.
What I am uncertain about is how to incorporate these big ideas into an already long and complicated text. As one of my reviewers noted, there are already a lot of big-concept keywords associated with this article. I wonder if you might pose one or two leading questions that might help me understand how you would like to see these elements of the pig-people relationship story brought into the paper.
Thank you for your close reading of my work. I am finding this entire review process very helpful to me as a thinker and a writer.